# Nonlinear and Synergistic Effects of Built Environment Indicators on Street Vitality: A Case Study of Humid and Hot Urban Cities

Jilong Li , Shiping Lin *, Niuniu Kong, Yilin Ke, Jie Zeng and Jiacheng Chen

College of Tropical Agriculture and Forestry, Hainan University, Haikou 570208, China; 22220953000029@hainanu.edu.cn (J.L.); 18889930215@163.com (N.K.); 13350732876@163.com (Y.K.); zengjie0520@163.com (J.Z.); 18840653845@163.com (J.C.)
* Correspondence: lsplxy@163.com

**Abstract:** Street vitality has become an important indicator for evaluating the attractiveness and potential for the sustainable development of urban neighborhoods. However, research on this topic may overestimate or underestimate the effects of different influencing factors, as most studies overlook the prevalent nonlinear and synergistic effects. This study takes the central urban districts of humid–hot cities in developing countries as an example, utilizing readily available big data sources such as Baidu Heat Map data, Baidu Map data, Baidu Building data, urban road network data, and Amap's Point of Interest (POI) data to construct a Gradient-Boosting Decision Tree (GBDT) model. This model reveals the nonlinear and synergistic effects of different built environment factors on street vitality. The study finds that (1) construction intensity plays a crucial role in the early stages of urban street development (with a contribution value of 0.71), and as the city matures, the role of diversity gradually becomes apparent (with the contribution value increasing from 0.03 to 0.08); (2) the built environment factors have nonlinear impacts on street vitality; for example, POI density has different thresholds in the three cities (300, 200, and 500); (3) there are significant synergistic effects between different dimensions and indicators of the built environment, such as when the POI density is high and integration exceeds 1.5, a positive synergistic effect is notable, whereas a negative synergistic effect occurs when POI is low. This article further discusses the practical implications of the research findings, providing nuanced and targeted policy suggestions for humid–hot cities at different stages of development.

**Keywords:** built environment; street vitality; threshold effect; synergistic effect; humid–hot climate; gradient boosted decision trees (GBDTs) model; multi-source data

## 1. Introduction

Since Jane Jacobs first introduced the concepts of urban vitality and street vitality, there has been widespread theoretical exploration and empirical research into urban vitality and its influencing factors [1,2]. However, whether these theories or empirical results are applicable to humid–hot cities in developing countries remains unknown, and collecting and quantifying the multidimensional factors of the built environment still poses a challenge [3]. Streets, as the main public space of urban life where people walk, socialize, work, shop, and seek entertainment, support the diversity and vitality of a city [4]. Vibrant urban spaces support diverse human activities, fostering social communication and interaction, thereby benefiting long-term sustainable development [5]. Similarly, as the material carriers of human activities, street spaces are closely linked to urban vitality [6–8]. Therefore, understanding the impact of the built environment on street vitality is necessary for city managers and urban planners [9].

Urban vitality can be seen as a social process closely related to the built environment [10]. Many empirical studies have explored the strong correlation between the built

environment and street vitality [11]. However, most scholars use traditional regression models, assuming linearity or predefined models, overlooking the generally present nonlinear and synergistic effects [12,13]. From this perspective, they may overestimate or underestimate the impact of certain factors in the built environment. Meanwhile, the combination of variables at different values could either amplify or diminish these impacts [14]. Wang Zimeng and others have already discovered the pervasive nonlinear relationship between built environment indicators and street vitality in the main urban areas of Wuhan, but they did not explore the synergistic effects among these indicators [15]. Wu and others have found that the threshold effect of built environment variables has more practical guidance for active travel [16,17].

This study aims to quantitatively explain the nonlinear and synergistic effects of various indicators of the street-built environment on street vitality. Based on multi-source open big data from the central urban areas of Haikou, Nanning, and Guangzhou, continuous one-week data from the Baidu Heat Map (or Baidu Huiyan) were collected as the representation data of street vitality. Jane Jacobs emphasized the importance of urban construction intensity and diversity for urban vitality and community interaction in "*The Death and Life of Great American Cities*", highlighting mixed-use, walkability, and community interaction in cities [18]. Jan Gehl advocates for accessibility and pedestrian-friendly urban planning [19]. Richard Florida proposed the concept of "Creative Cities", considering the diversity, cultural environment, and innovative capacity of cities as crucial for attracting talent and driving economic growth [20]. Jan Brueckner studied the impact of urban intensity and accessibility on the real estate market and housing choices, emphasizing their influence on urban economics and society [21]. The research and theories of these experts and scholars emphasize the importance of intensity, accessibility, and diversity in urban planning, contributing to creating more livable cities with social interaction and economic vitality. Thus, this paper analyzes the main characteristics of the street-built environment from four dimensions: construction intensity, diversity, functionality, and accessibility. Moreover, we used an advanced machine learning method to construct the model, namely Gradient Boosting Decision Trees (GBDTs), and we applied the SHapley Additive exPlanations (SHAP) model to interpret the GBDT model, which is very helpful in explaining the decision-making process and features the importance of the model. This approach has been widely used in the data science and machine learning community [22–24]. Based on game theory, SHAP can provide the relative importance of each factor in the built environment, local explanations, and the interaction effects between factors. Based on this, we can provide some scientific suggestions for creating more vibrant streets. The contributions of this study are as follows: (1) it demonstrates the nonlinear effects of various factors in the built environment on street vitality; (2) quantitatively analyzes the synergistic effects between different indicators of the street-built environment; and (3) provides nuanced recommendations for enhancing street vitality in humid–hot cities at different development stages in developing countries.

### 1.1. Definition and Quantification of Street Vitality

The concept of "vitality" originates from biology, characterizing the capacity for vigorous life and sustained development [25]. Jane Jacobs believed that the diversity of urban life fosters urban vitality; Jan Gehl pointed out that street vitality stems from slow traffic. For a long time, research on streets in China from the perspective of architectural design and urban planning primarily focused on visual aesthetics, spatial form, and traffic efficiency, often overlooking human activities on the streets. A street's vitality depends on the presence and activities of people; the activities of people on a street are the source of its vitality, and the essence of street vitality lies in the diverse activities engaged by people on the street. There have been two prevailing views in explaining the concept of urban vitality: urban sociology and architecture [26]. Urban sociology generally believes that economic, social, and cultural vitality are intertwined with urban vitality. Urban vitality is the spatial representation of economic, social, and cultural activities [27]. Conversely,

architects consider urban spatial vitality as a form of urban activity based on urban spatial form, which can be created through design [28,29].

From the existing descriptions of street vitality, it is clear that there is no definitive definition yet [10], but street vitality can be understood as the interaction between people, their behaviors, and the physical space of the street [10,30]. The behaviors of people in street spaces create corresponding social, ecological, and economic benefits. The behavioral activities of people in the physical space of the street are considered the external representation of street vitality, which are reflected in two dimensions: time and space. Temporally, it is represented by the variability in people's activities at different times and their duration; spatially, it is demonstrated by the mobility and density of people within the spatial carrier. In this paper, street vitality primarily focuses on its social aspect, related to the characteristics of street space. It can be measured by examining the density of people engaged in a series of activities on the streets [30].

### 1.2. The Impact of the Built Environment on Street Vitality

Sociologists and architects have proposed various theories on how the urban built environment promotes urban vitality. Jane Jacobs emphasized the importance of diversity, density, and mixed-use in cities for urban vitality, as well as pedestrian-friendly urban design, street hierarchy, community involvement, and the significance of urban public spaces [30]. Jan Gehl focused on humanistic urban design, advocating for creating urban environments where people want to live, work, and play. His theory underscores urban sustainability and walkability [19]. Kevin Lynch highlighted that urban vitality is related to diversity, social connections, and openness within a city. He focused on social interactions and collaborations in cities, influencing research in urban sociology and urban planning, and emphasized the close link between urban vitality and social interaction [31]. Montgomery focused on the historical, cultural, and architectural aspects of cities, emphasizing the impact of cultural and social backgrounds on urban vitality [2].

These qualitative studies have inspired a series of quantitative research. For instance, many studies have empirically tested the theories of Jacobs and others in non-American contexts. Liu Mei and Li Qian's research suggests that increasing the social function density and the mix of social functions on streets might be more realistic for enhancing street vitality [32,33]. However, Wu and others noted that for high-intensity areas, increasing the mix of building functions and the intensity of surrounding blocks does not necessarily enhance street vitality, and this was not further explored in their study [34].

Moreover, with the advent of big data, there has been a significant increase in research related to the built environment and urban vitality in recent years. For instance, Wang Bo, Zhong Weijing, and their colleagues have analyzed the spatio-temporal dynamics of urban vitality using 1 km × 1 km grid cells and 2-h time intervals. They found that the impact of the built environment on street vitality exhibits patio-temporal heterogeneity [35,36]. Tana and colleagues, utilizing big data, investigated the relationship between the built environment and urban vitality in central Shanghai. Their findings suggest that the influence of block-scale built environment features on vitality varies. They observed that increased POI density, road network density, and diversity of POIs enhance urban vitality. Conversely, a higher number of building floors and greater building density tend to decrease social vitality while bolstering economic vitality in blocks. Additionally, improved transport accessibility is beneficial for economic vitality [37]. However, Yun Han, employing machine learning methods, discovered that high accessibility around streets with low functional diversity might lead more to external travel than to local social activities. He also highlighted that due to interactions between the built and socioeconomic environments, similar studies show considerable variation across different cities. Therefore, planning practices should meticulously consider local environment-related thresholds [38].

## 2. Conceptual Framework

Urban street vitality is a pivotal aspect of urban development, necessitating a deep understanding and precise calibration of four essential dimensions of the built environment: construction intensity, diversity, functional nature, and accessibility. Construction intensity, defined as the density of population and buildings, engenders a bustling, diverse urban milieu, thereby invigorating urban streets [38]. Diversity, encompassing cultural, commercial, social, and economic facets, enriches urban streets by offering a variety of activities and choices [39]. The functional nature signifies the primary purposes and characteristics of a street. Meanwhile, accessibility, as another critical dimension, ensures that urban centers of activity are easily and efficiently reachable, fostering social interactions and economic undertakings on urban streets [40,41]. Through meticulous measurement and harmonization of these four dimensions' key indicators, vibrant, socially engaging, and appealing urban streets can be realized. This not only furthers the sustainable development of cities but also elevates the quality of life for residents, thereby shaping the city's unique identity and attractiveness.

Jane Jacobs, in her seminal work *"The Death and Life of Great American Cities"*, emphasized the critical role of high construction intensity in urban areas. She contended that substantial population and building density are necessary to sustain vibrant street life and social interactions. Jacobs was a proponent of mixed-use buildings and advocated for small-block urban designs, arguing that these elements promote construction intensity, thus contributing to the vibrancy of cities. Additionally, Jacobs highlighted the significance of urban diversity, advocating for a blend of cultural, commercial, and social activities. She believed this mix attracts diverse groups of people, fostering social interaction. Her opposition to monofunctional urban planning was rooted in her support for a combination of mixed-use developments and multiculturalism [18]. Jan Gehl, another influential urbanist, champions the importance of diversity and functionality in streets. His work advocates for multifunctional land use, embracing multiculturalism, and enriching public spaces. Gehl argues that cities should cater to a broad spectrum of cultural, entertainment, commercial, and social activities, meeting the diverse needs of their inhabitants and thereby enhancing social interaction and urban vitality. He also emphasizes the necessity of accessible urban environments, particularly focusing on walkability and cyclability. Gehl is a strong advocate for pedestrian-friendly urban design in central areas, emphasizing well-designed sidewalks, intersections, and public transit systems. These elements facilitate easier access to urban centers, promoting social engagement and vitality [19]. The synergistic effect of these four dimensions—construction intensity, diversity, functionality, and accessibility—is crucial for fostering urban street vitality. This interplay warrants deeper exploration and research to fully understand and enhance the dynamics of urban life.

## 3. Research Area and Data Acquisition

### 3.1. Study Area

The humid subtropical region defined in this paper is classified based on the system developed by the German climatologist Wladimir Peter Köppen, primarily using temperature and precipitation as key indicators and incorporating the Köppen climate classification [42]. Dr. Haiyan Yan from Xi'an University of Architecture and Technology further utilized typical temperature and humidity combinations in winter and summer as the classification criteria, designating cities such as Guangzhou, Nanning, and Haikou in China as representatives of the temperate–humid subtropical region. This translation is intended for use in a scientific paper submission and emphasizes precision and accuracy [43].

Haikou, Nanning, and Guangzhou, provincial capitals in Southern China, each hold distinct roles in the country's urban landscape. Haikou, a budding free trade port city, is poised to become a new economic growth engine, exemplifying the fusion of economy and ecology. Nanning serves as a transitional city, linking China's past and future. Guangzhou, recognized as a significant central city and a global first-tier city by the State Council, plays a crucial international role. The three cities mentioned are all located in the humid subtropical

region of southern China, sharing similar climatic conditions. Their street vitality formation mechanisms exhibit similarities as well. They serve as China's gateways to Southeast Asian cities. By exploring the mechanisms and subtle patterns underlying the formation of street vitality in these cities, we can further uncover the crucial relationship between urban development and economic growth in developing countries. This research can provide valuable insights into the construction and development of cities in Belt and Road Initiative countries, such as Southeast Asia. Please note that this translation is intended for use in a scientific paper submission and is aimed at precision and clarity.

According to the "2023 City Business Attractiveness Ranking" by the First Finance and Economics New First-Tier City Research Institute, Guangzhou, Nanning, and Haikou epitomize China's first-tier, second-tier, and third-tier cities, respectively. This ranking, based on factors like city size, population, and economic vitality, positions Guangzhou at the top, followed by Nanning and Haikou (Table 1). Consequently, this study selects Haikou as an emblematic early-stage development city in China's humid and hot region, Nanning as a city in the midst of development, and Guangzhou as a more mature city.

**Table 1.** The comparative data of three cities.

| City | Built-Up Area (km$^2$) | Year-End Permanent Resident Population (Thousands) | Gross Regional Product (CNY Billion) |
|------|------------------------|---------------------------------------------------|--------------------------------------|
| Haikou | 165.20 | 293.97 | 2134.77 |
| Nanning | 319.69 | 889.17 | 5218.34 |
| Guangzhou | 1380.60 | 1873.41 | 28,839.00 |

Finally, after an initial assessment using Baidu Heat Maps, the study areas were delineated as follows: for Haikou City, the central high-vitality area extends from Haidian Island in the north to the Ring Expressway in the south, from Old Town Station in the west to the Nandu River in the east, encompassing the coordinates 19°55′54.21″ N to 20°05′48.87″ N and 110°08′18.03″ E to 110°23′28.47″ E. In Guangzhou City, the central urban area is defined within 22°52′14.20″ N to 23°18′03.86″ N and 113°09′52.97″ E to 113°33′28.68″ E. For Nanning City, it is outlined between 22°39′44.13″ N to 22°54′22.45″ N and 108°26′09.45″ E to 108°08′49.82″ E. The road networks of these study areas are depicted in Figure 1.

*3.2. Data Acquisition*

Acquisition and Processing of Multi-Source Data:

(a)  The road network data were sourced from Baidu Maps and mapped using ArcMap 10.8. Streets of different levels were identified and assigned unique attributes and identifiers, resulting in a comprehensive vector road network database containing various fields like length, street level, and identifier.

(b)  Baidu Heat Map data (Baidu Huiyan) were acquired via Python web scraping techniques. To reduce random errors, a continuous dataset of one-week heat map TIFF images for Haikou's central urban area was collected (from 24 December to 30 December 2022, from 05:00 AM to 11:00 PM, with an hourly frequency, totaling 126 images). A similar dataset was gathered for the same area from 10 July to 16 July 2023. These data were processed in ArcMap 10.8 to calculate the average heat values for each street segment in the study area over a week.

(c)  Amap (Gaode) POI data were obtained through an official paid service. The data for Haikou were collected in April 2023, for Guangzhou in July 2023, and for Nanning in October 2023. To ensure data freshness, the interval between acquiring the POI data and the heat map data was kept under six months.

(d)  Building contour data from Amap (Gaode), including building heights, were sourced from Shuijing Weitu and updated in 2022 for Haikou, Nanning, and Guangzhou.

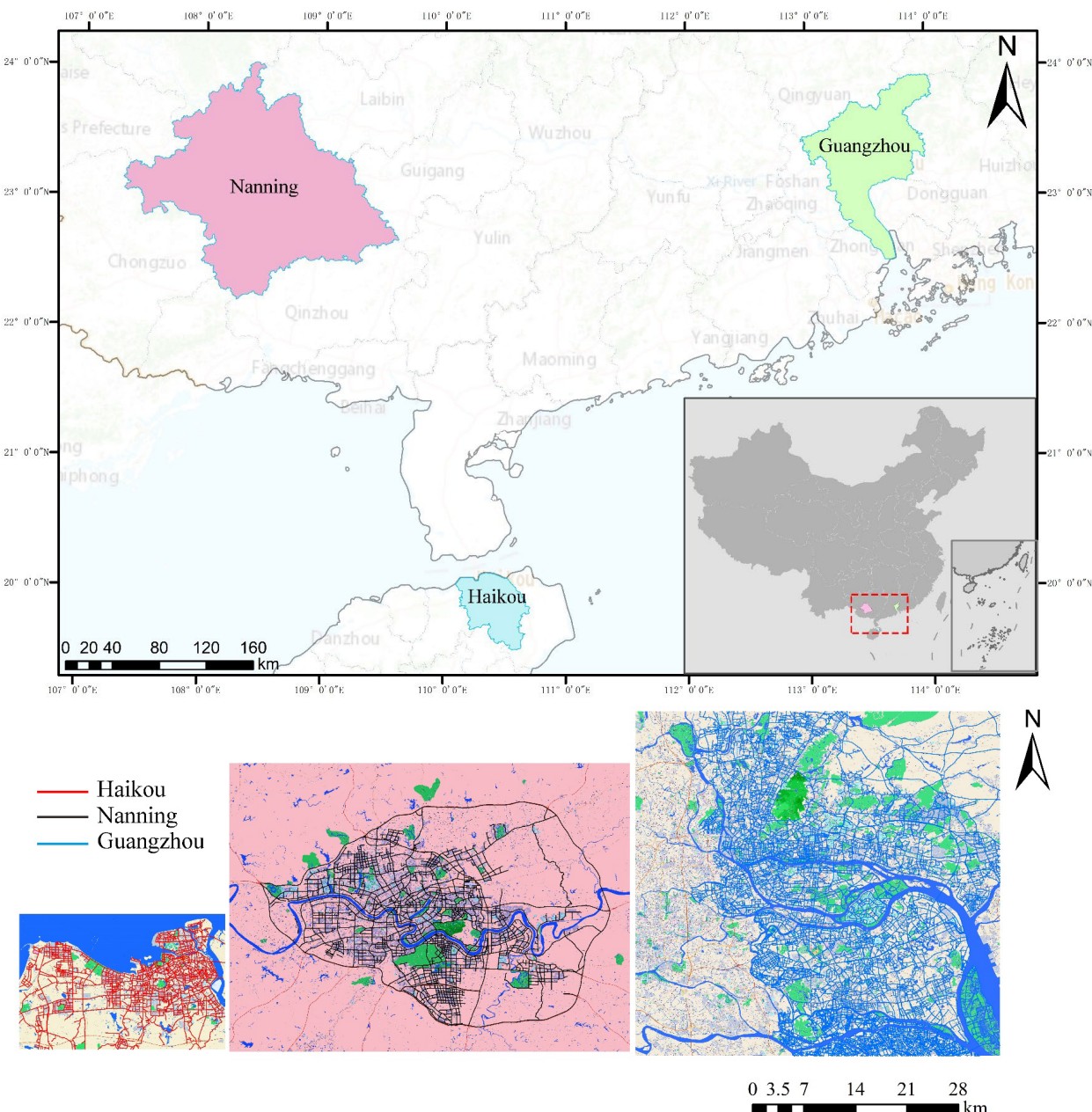

**Figure 1.** Study area.

## 4. Methods

A methodological framework has been established to investigate the nonlinear and synergistic impacts of various built environment indicators on street vitality. As depicted in Figure 2, this framework unfolds in four stages: (1) quantitative acquisition of thermal data for each street segment in the three cities, utilizing Baidu Heat Map and Location-Based Services (LBSs) data as proxies for external manifestations of street vitality; (2) collection of 11 built environment indicators from Gaode building data and Baidu POI data, serving as independent variables in the model; (3) the application of the GBDT modeling technique to delineate the relationship between each built environment indicator and street vitality; and (4) the utilization of the SHAP model to interpret the GBDT model, enabling a deeper analysis of how built environment indicators in cities at different stages of development, especially in hot and humid climates, influence street vitality, as elaborated in the subsequent sections.

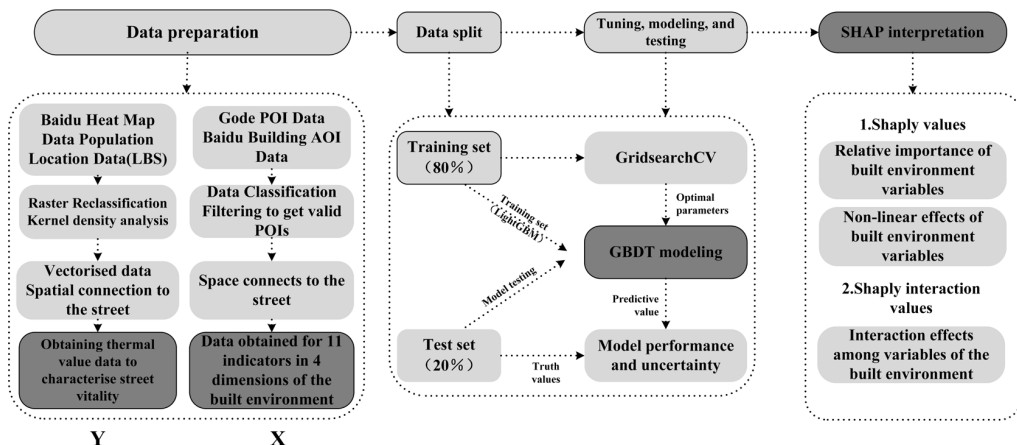

**Figure 2.** Methodological framework.

*4.1. Variables*

Traditional research on street vitality often depends on field surveys, questionnaires, and interviews for data collection. This approach's strength lies in its comprehensive nature, allowing for the acquisition of specific, tailored information under researcher supervision, thereby ensuring accuracy. This method is particularly effective for micro-scale street studies. However, such data are inherently static, capturing the characteristics of specific streets at discrete moments. Furthermore, due to resource constraints, it is challenging to extend this approach to large-scale, cross-regional studies. In the information age, the advent of LBSs and Baidu's heat map data have mitigated these limitations. Baidu's population heat data, derived from the activities of millions of Baidu app users, offers extensive spatial and temporal coverage [44,45]. These data enable a more robust quantitative analysis of the complex, synergistic impacts of built environment factors on street vitality. Utilizing ArcMap 10.8 for the quantitative analysis of this big data, the average vitality of each street segment in the study areas over a continuous week is ascertainable, as detailed in Table 2.

**Table 2.** Dependent variables of street vitality in three cities.

| City | Data Content | Description | Source | Max | Min | Mean | Sta |
|---|---|---|---|---|---|---|---|
| Haikou | Baidu Heat Map Grid Data | By vectorising 140 heat map grid images and by processing Baidu's population LBS data at 80 time points, the average heat value of the corresponding streets is obtained | a | 4.83 | 0.30 | 2.69 | 1.05 |
| Nanning | Baidu Huiyan LBS Data | By processing Baidu's population LBS data at 80 time points, the average heat value of the corresponding streets is obtained | b | 7.47 | 1.16 | 4.16 | 1.12 |
| Guangzhou | Baidu Huiyan LBS Data | By processing Baidu's population LBS data at 80 time points, the average heat value of the corresponding streets is obtained | b | 7.79 | 1.00 | 3.27 | 0.96 |

Data sources: (a) Baidu Maps App; (b) Baidu Huiyan Maps.

This study establishes eleven indicators to assess the built environment of urban streets, encompassing four key dimensions (Table 3). For construction intensity, four indicators are utilized: POI density, number of buildings, building height, and building footprint area.

**Table 3.** Eleven indicators of street-built environment across four dimensions.

| Dimension | Indicator | Description | Source | Mean (Haikou/ Nanning/ Guangzhou) | Std (Haikou/ Nanning/ Guangzhou) |
|---|---|---|---|---|---|
| Construction intensity | POI density | Linear density of the number of POIs within the range of each street segment (per km) | a | 221.01/ 156.83/ 225.27 | 181.02/ 128.54/ 169.27 |
| | Number of buildings | Linear density of the number of buildings within the range of each street segment (per km) | a | 108.78/ 79.18/ 77.49 | 73.52/ 57.15/ 58.72 |
| | Building perimeter | Total perimeter of all buildings within the range of each street segment divided by street length (m/km) | a | 10,887.96/ 6003.80/ 7535.20 | 6282.15/ 3178.60/ 3947.95 |
| | Building footprint area | Total area of building occupation within the range of each street segment divided by street length (m$^2$/km) | a | 58,638.65/ 250.90/ 40,366.44 | 35,694.24/ 130.40/ 25,871.39 |
| | Building height | Total height of all buildings within the range of each street segment divided by street length (m/km) | b, c | 5110.82/ 2316.21/ 1516.46 | 3646.70/ 1694.71/ 1518.69 |
| Diversity | Diversity | Shannon diversity index (the larger the value, the richer the diversity) | a | 2.00/2.04/1.93 | 0.31/0.30/0.36 |
| Functional Properties | Recreational facility density | Number of recreational facilities divided by street length | a | 16.40/10.98/ 14.48 | 16.39/10.66/ 13.78 |
| | Residential facility density | Number of residential facilities divided by street length | a | 15.90/7.40/ 12.22 | 15.19/8.99/ 12.04 |
| | Office facility density | Number of office facilities divided by street length | a | 23.92/15.27/ 29.22 | 28.37/18.07/ 30.29 |
| Accessibility | choice | Standardized angular choice through Depth map, formula = log((choice (radius)) + 1) divided by log((total depth (same radius as numerator)) + 3) | d | 0.94/0.89/0.90 | 0.18/0.17/0.19 |
| | Integration | Standardized angular integration through Depth map, formula = 1.2th 2nd power of node count divided by (total depth + 2) | d | 0.09/0.11/0.09 | 0.01/0.01/0.14 |

Data source:(a) Amap open platform; (b) Baidu Map Open Platform; (c) Water Economic Map; (d) Baidu Map.

(a)  POI density indicates the concentration of functional points in an area, with high values suggesting a rich and vibrant urban functionality, reflecting urban development intensity [46]. The number of buildings quantitatively represents the construction volume in an area, often correlating with economic growth and population density, thereby serving as a direct measure of urban construction intensity.

(b)  Building height serves as an indicator of urban modernization and vertical expansion. High-rise structures typically denote business hubs and dense residential zones, signifying efficient urban space usage and construction intensity. Building footprint area measures the land coverage by buildings, reflecting urban planning efficiency and land use in areas with limited land resources, thus contributing to the assessment of urban construction intensity. These four indicators collectively provide a multifaceted view of urban street construction, encompassing land use efficiency, building distribution and scale, and urban functionality.

(c)  Diversity is measured using the Shannon Index, a concept from biology, to evaluate the variety and distribution uniformity of the urban street ecosystem.

(d)  Functional Nature is assessed through the density of recreational, residential, and office facilities. Recreational facility density indicates the ratio of leisure-oriented amenities to the length of a street segment, reflecting the street's role in leisure and cultural life. Residential facility density, showing the concentration of living spaces, indicates the residential aspect of the street. Office facility density, denoting the pres-

ence of commercial and business spaces, reflects the street's economic function. These indicators collectively shed light on the primary usage and functional characteristics of urban spaces.

(e) For connectivity, two indicators, choice and integration, are chosen. Choice measures the availability of different routes from a point, indicating the ease of reaching various destinations. Integration reflects a point's centrality within the network, with high values suggesting better accessibility. Enhanced choice and integration potentially increase pedestrian and traffic flow, thereby boosting street vitality and appeal.

### 4.2. Modeling Approach

The GBDT is an ensemble learning method combining decision tree algorithms with gradient boosting. GBDT utilizes a sequence of decision trees as base models [47,48], each functioning as a weak learner that improves predictive accuracy iteratively. In this study, model development utilized the LightGBM and Scikit-learn packages in Python 3.7. The dataset was randomly divided into a training set (80% of the data) and a test set (20%). Mean Squared Error (MSE) served as the loss function. To enhance model performance and mitigate overfitting, Grid Search with 5-fold cross-validation (GridSearchCV) was employed for hyperparameter tuning, resulting in the optimal parameter set (detailed in Table 4). Subsequently, the model underwent sequential training. For a thorough evaluation, various metrics such as Mean Absolute Error (MAE), Root Mean Squared Error (RMSE), and the R-squared value were applied. Moreover, an ensemble-based approach was used to assess the model's predictive uncertainty.

**Table 4.** Parameters specified in this study.

| Hyper-Parameters | Descriptions | Optimal Hyperparameters | | |
| --- | --- | --- | --- | --- |
| | | Haikou | Nanning | Guangzhou |
| Num_Leaves | Determines the number of leaves in each tree | 21 | 15 | 29 |
| Learning_Rate | The magnitude of model parameter updates in each iteration | 0.004 | 0.001 | 0.001 |
| Feature_Fraction | Specifies the proportion of features considered when splitting each tree node | 0.707 | 0.769 | 0.773 |
| Bagging_Fraction | Determines the fraction of data used for training in each iteration | 0.727 | 0.761 | 0.697 |
| Bagging_Freq | Specifies how often data sampling is performed during training | 10 | 2 | 8 |
| Max_Depth | Limits the depth of each tree to prevent overfitting | 5 | 12 | 5 |
| Min_Child_Weight | Defines the minimum sum of sample weights required in a leaf node | 2.758 | 3.546 | 9.911 |
| Num_Boost_Round | Specifies the number of boosting rounds or trees in the ensemble | 1071 | 4495 | 7997 |

Local explanation methods play a crucial role in providing targeted explanations for individual predictions, enhancing our understanding of the intricate, nonlinear interactions between various components of the built environment and street vitality. SHAP, a key local interpretative method, is grounded in the principles of classic cooperative game theory and involves computing Shapley values [49,50]. In the context of machine learning, predictions can be somewhat likened to a game, where each prediction is influenced by a combination of variables. Shapley values distribute the 'credit' or 'impact' of explanatory variables fairly by assessing their average marginal contribution across all conceivable combinations of

variables. Essentially, Shapley values provide an equitable and detailed method to account for the influence of each variable in the prediction process:

$$\Phi_k(f,x) = \sum_{s\in\phi} \frac{1}{K!}\left[f_x\left(P_k^s\bigcup k\right) - f_x(P_k^s)\right] \tag{1}$$

In this context, $\Phi_k(f,x)$ represents the Shapley value, which reflects the average impact of variable $k$ compared to the overall mean prediction on an individual prediction. $s$ is the set of possible permutations of variables, and $K$ represents the total number of variables. In the permutation $s$, the set of variables preceding variable $k$ is considered, and $x$ is the value of the explanatory variable. Subsequently, the $f(x)$ explanation for an individual prediction can be provided as follows:

$$f(x) = \Phi_o(f,x) + \sum_{k=1}^{k} \Phi_k(f,x) \tag{2}$$

In this context, $\Phi_o(f,x)$ represents the overall predicted mean of the population. The relative importance of variables is computed by averaging the absolute Shapley values for each variable:

$$I_k = \frac{1}{n}\sum_{i=1}^{n}\left|\Phi_k^{(i)}\right| \tag{3}$$

In this context, $I_k$ represents the importance of variable $k$ and $\Phi_k^{(i)}$ denotes the Shapley value of variable $k$ for a single prediction $i$. In essence, Shapley values can estimate the local impact of explanatory variables on the final prediction. The local impact of explanatory variables (i.e., Shapley values) can be further decomposed into their primary local effects and local interaction effects with other variables (i.e., Shapley interaction values). Shapley interaction values capture local interaction effects by attributing credit among variable pairs. The definition of Shapley interaction values is

$$\Phi_{i,j}(f,x) = \sum_{T\in\gamma\setminus\{i,j\}} \frac{|T|!(K-|T|-2)!}{2(K-1)!}\nabla_{i,j}(f,x,T) \tag{4}$$

When $i \neq j$,

$$\nabla_{i,j}(f,x,T) = f_x(T\cup\{i,j\}) - f_x(T\cup\{i\}) - f_x(T\cup\{j\}) + (T) \tag{5}$$

In this context, $\Phi_{i,j}(f,x)$ represents the Shapley interaction value, reflecting the interaction effects between variables and variable $i$ in a single prediction. $K$ is the number of variables, $x$ is the set of input variables, and $T$ is the set of potential variables.

## 5. Results and Discussion

### 5.1. Relative Importance of Indicators

We developed a GBDT model utilizing LightGBM, and applied SHAP for in-depth model interpretation. The model incorporated various elements of the urban built environment as input variables, with street thermal data functioning as the response variable. Detailed performance metrics of the model are presented in Table 5. To ensure the model's reliability and stability, K-fold cross-validation was conducted. As depicted in Figure 3, we obtained the Precision–Tradeoff (P–T) curve for the refined model, demonstrating its predictive capability.

**Table 5.** Model performances.

| | | MSE | | RMSE | | MAE | | R-Squared | |
|---|---|---|---|---|---|---|---|---|---|
| | | **Training Set** | **Test Set** | **Training Set** | **Test Set** | **Training Set** | **Test Set** | **Training Set** | **Test Set** |
| GBDT (lightGBM) | Haikou | 0.204 | 0.414 | 0.452 | 0.643 | 0.341 | 0.487 | 0.858 | 0.749 |
| | Nanning | 0.334 | 0.486 | 0.578 | 0.697 | 0.425 | 0.516 | 0.792 | 0.755 |
| | Guangzhou | 0.241 | 0.352 | 0.491 | 0.594 | 0.374 | 0.448 | 0.741 | 0.681 |

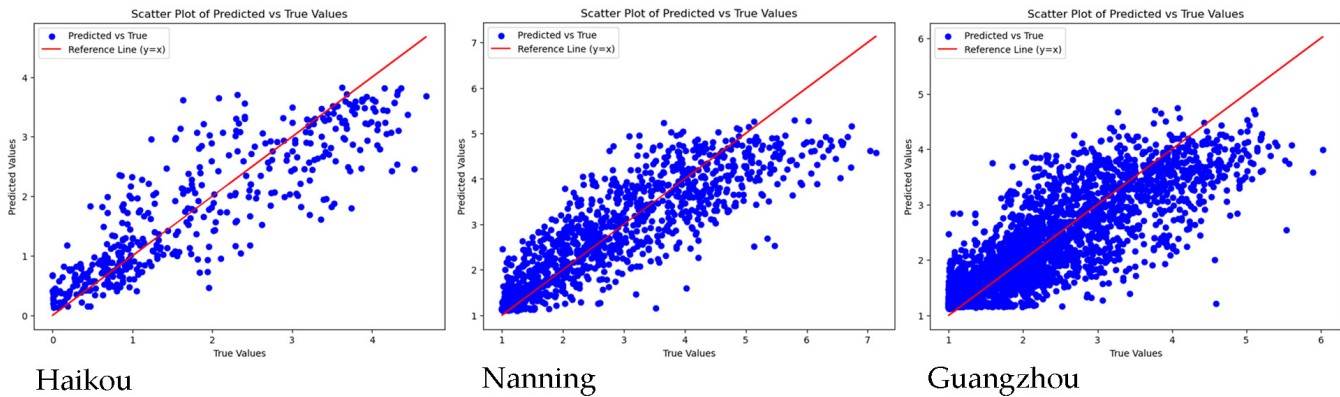

**Figure 3.** P–T (predicted value and true value) plots corresponding to models in three cities.

Figure 4 presents the relative significance of different variables in the built environment. On the left side, the variables are ranked in descending order of their overall importance. The right side of the figure demonstrates the contribution of each variable's values across different Metropolitan Statistical Areas (MSA) to street vitality. This global importance is determined by computing the average of the absolute Shapley values for each variable.

Table 6 summarizes the contribution values of each dimension for different cities. In the construction intensity indicator, Haikou shows the most significant contribution (SHAP value of 0.18), reflecting the typical necessity for extensive construction and infrastructure development in the initial stages of urban growth. Guangzhou, as a more mature city, still exhibits a high contribution in construction intensity (SHAP value of 0.17), suggesting ongoing infrastructure and growth demands. Nanning, with a lower SHAP value of 0.15 in construction intensity, indicates a lesser dependence on construction in its urban development strategy, possibly focusing more on other aspects.

**Table 6.** Importance data for 11 indicators across 4 dimensions.

| Dimension | Indicator | Haikou | | Nanning | | Guangzhou | |
|---|---|---|---|---|---|---|---|
| Functional density | POI density | 0.31 | | 0.37 | | 0.43 | |
| | Number of buildings | 0.33 | | 0.13 | | 0.16 | |
| | Building perimeter | 0.02 | 0.71 | 0.01 | 0.58 | 0.02 | 0.69 |
| | Building footprint area | 0.01 | | 0.03 | | 0.02 | |
| | Building height | 0.03 | | 0.03 | | 0.05 | |
| Diversity | Diversity | 0.03 | 0.03 | 0.07 | 0.07 | 0.08 | 0.08 |
| Facility density | Recreational facility density | 0.07 | | 0.06 | | 0.02 | |
| | Residential facility density | 0.06 | 0.18 | 0.05 | 0.15 | 0.06 | 0.14 |
| | Office facility density | 0.05 | | 0.05 | | 0.07 | |
| Accessibility | Choice | 0.02 | 0.09 | 0.04 | 0.20 | 0.03 | 0.09 |
| | Integration | 0.07 | | 0.16 | | 0.06 | |

In the dimension of urban diversity, Guangzhou leads with the highest contribution (SHAP value of 0.08), reflecting its emphasis on a diverse range of urban functions and activities, a trait commonly seen in more mature urban environments. Nanning is close behind with a SHAP value of 0.07, indicating its ongoing efforts to foster urban diversity, albeit slightly less than Guangzhou. Haikou, with a SHAP value of 0.03, shows the lowest diversity contribution, aligning with its stage of early urban development where a narrower range of functions might be prevalent.

In the dimension of functional nature, the SHAP values for Haikou, Nanning, and Guangzhou are relatively uniform, hovering around 0.05. This uniformity indicates that these cities similarly prioritize the distribution of diverse functions without a predominant focus on any specific type.

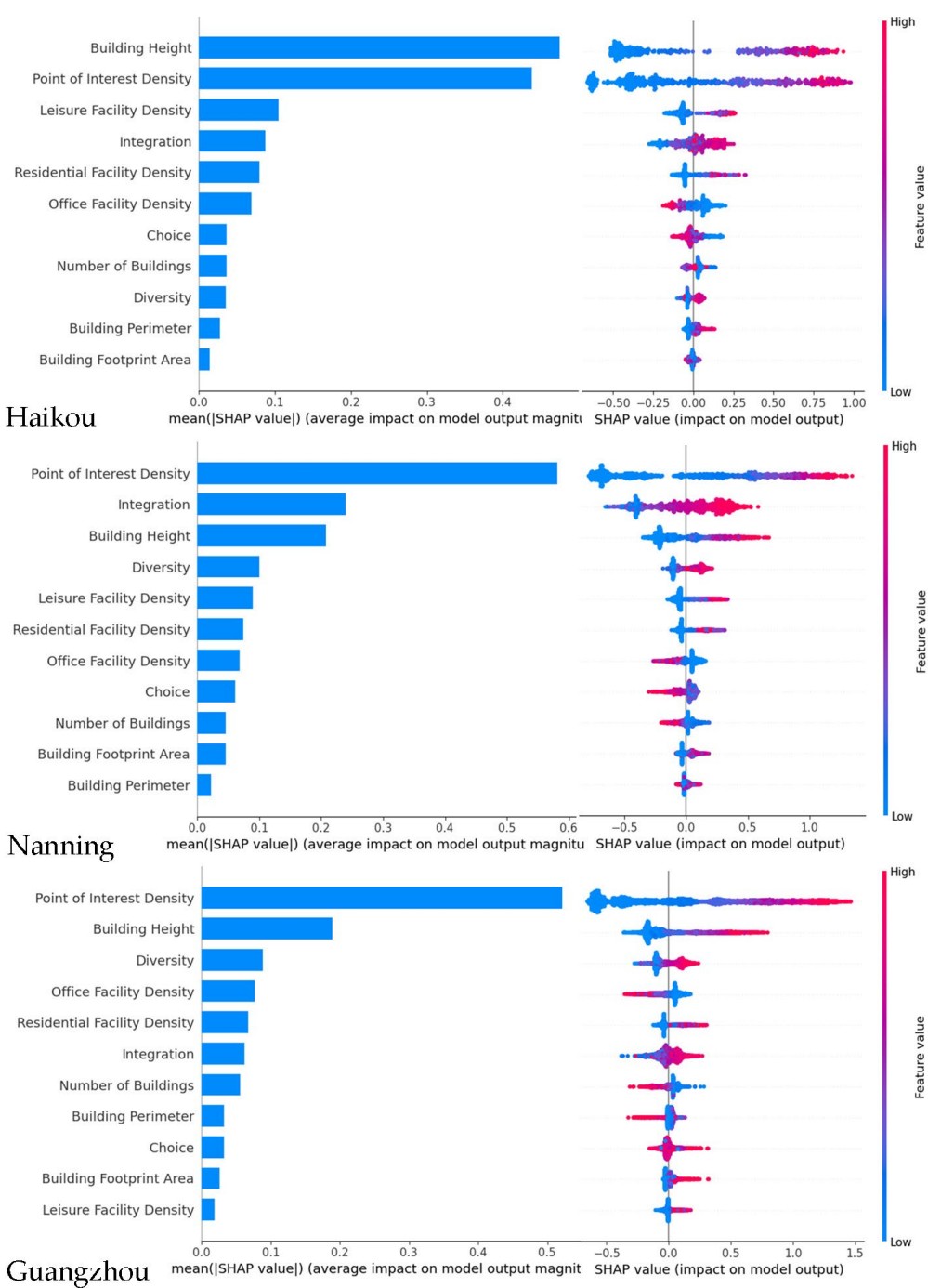

**Figure 4.** Ranking of indicator importance based on SHAP.

In terms of accessibility, Nanning leads with the highest contribution (SHAP value of 0.10), which may be attributed to its ongoing efforts in enhancing transportation and connectivity, reflecting a characteristic of mid-stage urban development. Haikou, still in its early phase of urban development, ranks second (SHAP value of 0.05) in accessibility contribution, indicating progressive improvements in this area. Conversely, Guangzhou, a more mature city, has the lowest accessibility contribution (SHAP value of 0.04), likely impacted by challenges such as traffic congestion, a common issue in well-developed urban areas.

### 5.2. The Nonlinear Effects of Street Built Environment

As illustrated in Figures 5 and 6, each local dependency plot corresponds to an indicator of the built environment, showcasing how that particular indicator influences street vitality. In each plot, every point represents a MSA, with the x-axis indicating the variable's value and the y-axis depicting its local effect on vitality. A comprehensive analysis of these plots reveals the following:

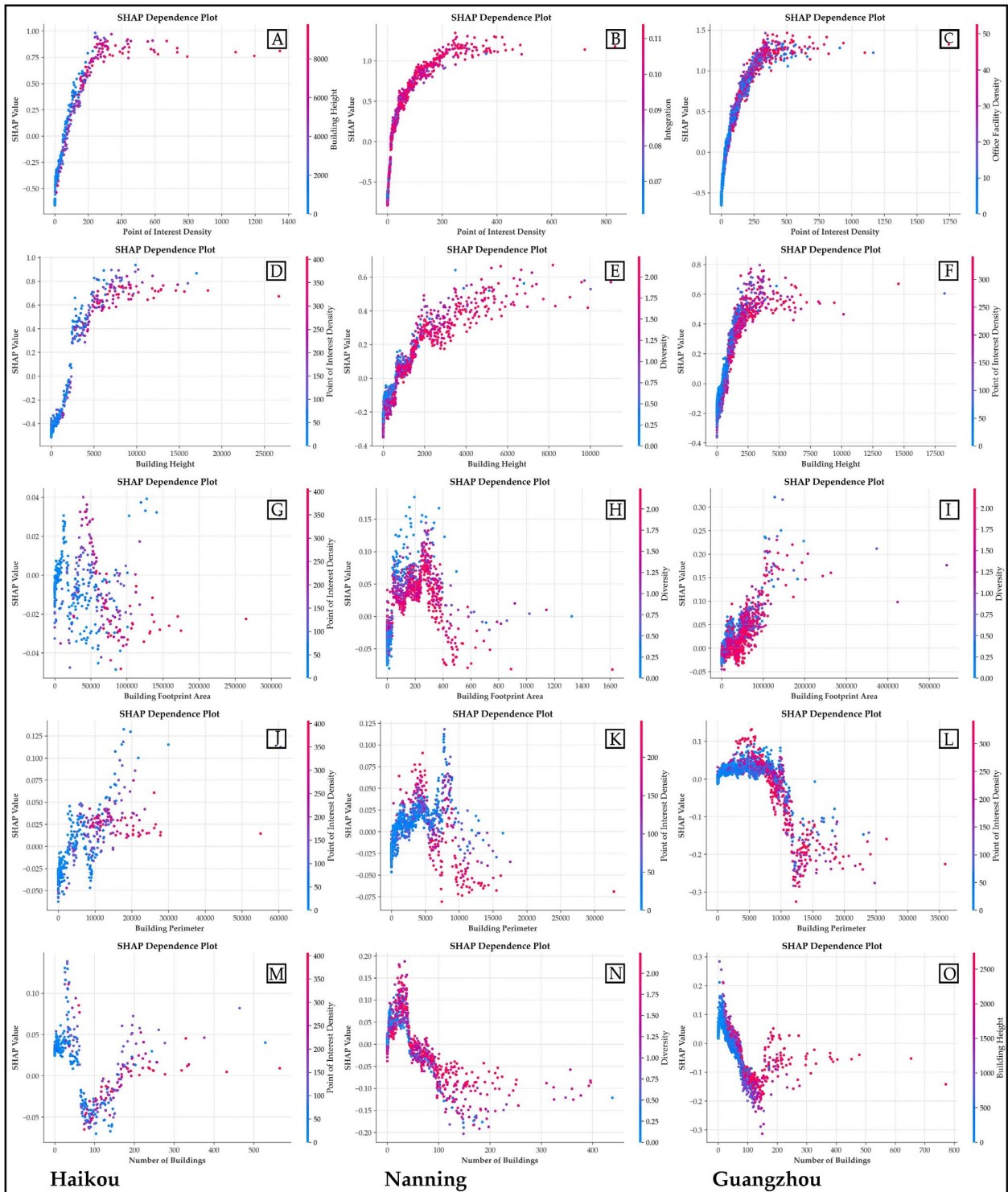

**Figure 5.** Map of local effects of built environment indicators (indicators 1–5).

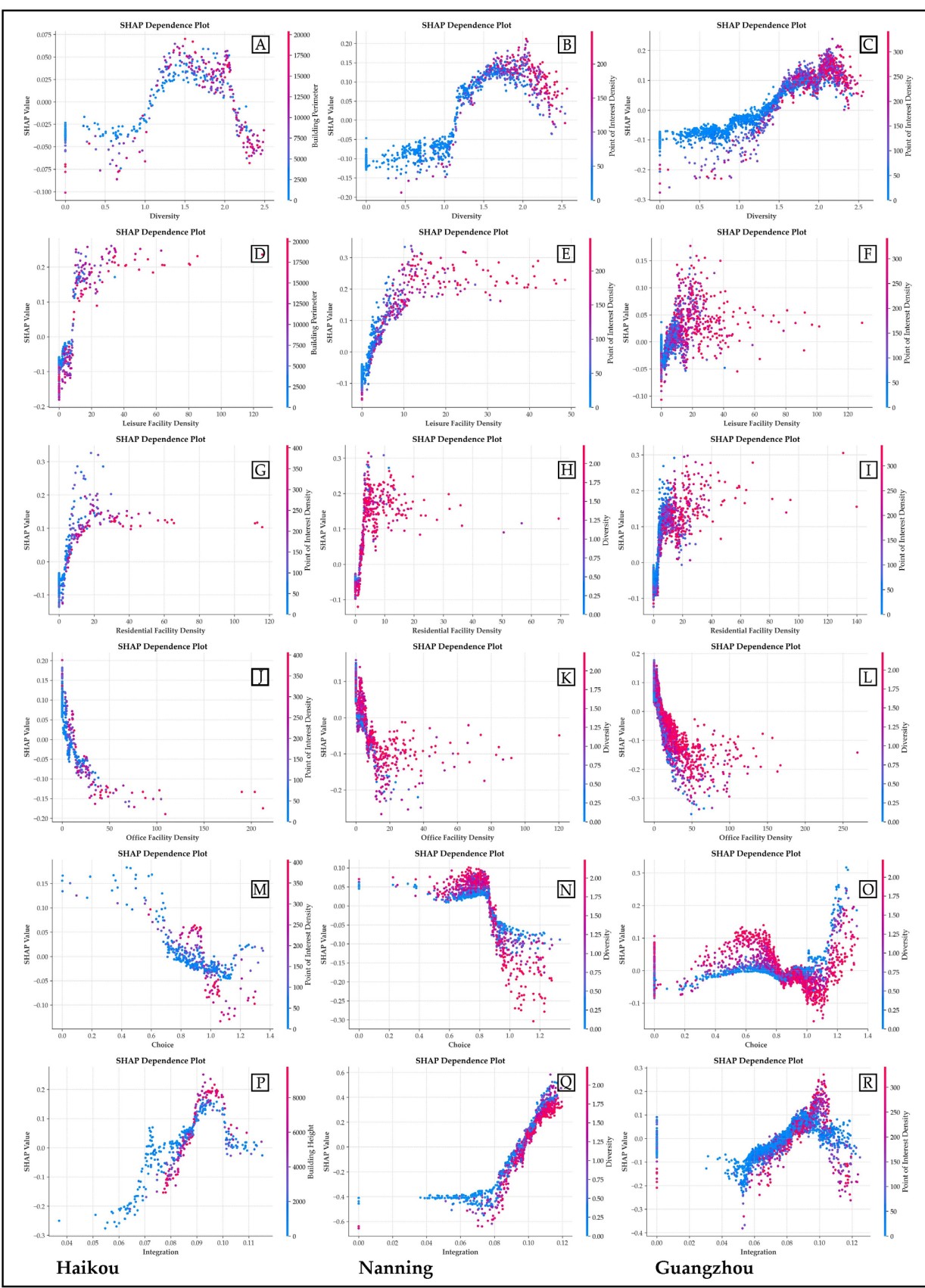

**Figure 6.** Map of local effects of built environment indicators (Indicator 6–Indicator 11).

Figure 5A illustrates the relationship between social function density and street vitality in Haikou. Initially, an increase in social function density correlates positively and sharply with street vitality's SHAP value. However, beyond a moderate density level, this contribution to vitality saturates and slightly declines, suggesting potential negative impacts of over-densification. In Figure 5B, Nanning's relationship between social function density and street vitality is depicted with a flatter positive trend. The peak SHAP values are noted at lower densities compared to Haikou. After reaching this peak, a significant decrease in SHAP values is observed as social function density continues to rise, indicating that excessively high density might adversely affect street vitality. Figure 5C demonstrates a consistently strong positive correlation between social function density and street vitality in Guangzhou. Here, the SHAP values stabilize following a rapid ascent without declining, implying that Guangzhou's higher developmental stage allows it to maintain or enhance street vitality even at higher densities.

This suggests that as cities develop from Haikou to Guangzhou, they become more adept at handling higher densities of social functions without negatively impacting street vitality. This capability is likely due to the advanced urban planning, infrastructure, and social services in more developed cities.

Figure 5D (Haikou) shows that the impact of building height density on street vitality shifts from negative to positive with increasing density. Considering Haikou's developing status, the initial increase in building density might not be accompanied by necessary infrastructure and social service enhancements, leading to vitality impairment. However, beyond a certain threshold, increased height density could bring more commercial and social activities, thus enhancing street vitality. Figure 5E (Nanning) displays a more linear relationship, indicating that throughout the range of building height density, the contribution to street vitality remains positive. This could reflect a scenario where, in Nanning, increases in building height are typically coupled with improvements in infrastructure and social services, directly fostering an increase in street vitality. For Figure 5F (Guangzhou), we observe a sustained and substantial positive influence of building height density on street vitality, albeit with a slight decrease at very high densities.

In summary, the level of urban development influences the contribution of building height density to street vitality. Cities in the early stages of development, such as Haikou, need to ensure that increased building height density is accompanied by enhanced infrastructure and social services to avoid negative impacts on vitality. More developed cities like Guangzhou need to consider diminishing marginal effects while increasing building density, as well as how to maintain and enhance street vitality through other means, such as increasing public spaces and improving environmental quality, to achieve sustainable urban vitality and quality of life.

As depicted in Figure 5G (Haikou), there is initially a positive correlation between building footprint density and street vitality, but as density reaches a certain level, the growth in vitality slows and even slightly declines. This may indicate that beyond a certain point, increased density could negatively impact vitality. Figure 5H (Nanning) shows a more gradual relationship, indicating higher street vitality at lower densities but a rapid decline in vitality beyond a certain threshold.

This suggests that moderate building footprint density is crucial for street vitality. In the case of Figure 5I (Guangzhou), as a more developed city, the chart shows that even at high densities, street vitality can be maintained or enhanced. This may indicate a stronger capacity to accommodate high-density development.

For cities in the early development stage (represented by Haikou in Figure 5J), lower building perimeter density might not be sufficient to support a vibrant street life, but as building density increases, there is a significant improvement in street vitality. Too low building density may imply dispersed commercial, residential, and other functional areas, which is not conducive to concentrated social and economic activities. In cities with medium development levels (represented by Nanning in Figure 5K), moderate building perimeter density might already support more active social and economic interactions.

However, due to the rapid urbanization process these cities might be undergoing, the relationship between street vitality and building density could be more complex and variable. For instance, there might be transitional areas where new constructions could enhance vitality, but at the same time, some older areas might not see the expected increase in vitality due to over-density. In highly developed cities (represented by Guangzhou in Figure 5L), increasing building perimeter density initially may rapidly boost street vitality, as a greater concentration of population and business brings more interaction and economic opportunities. However, beyond a certain threshold, excessive building perimeter density could lead to congestion, environmental degradation, and potentially the displacement of small, community-oriented businesses due to high land prices, thereby diminishing street vitality.

Figure 5M (Haikou) shows that an increase in building number density seems to positively contribute to street vitality, but this positive impact tends to plateau after reaching a certain density. Figure 5N (Nanning) demonstrates that an initial increase in the number of buildings positively impacts street vitality, which then quickly diminishes and trends negatively. Figure 5O (Guangzhou) exhibits a fluctuating trend, where building number density initially negatively impacts, then positively influences, but soon declines, street vitality. In highly developed cities, building numbers might have reached saturation, and street vitality is more influenced by other factors, such as building quality, service facilities, and public space configuration.

This indicates that in cities like Haikou, which are in the early stages of development, increasing the number of buildings can effectively enhance street usage and commercial activities. In cities like Nanning, in the middle stage of development, excessive buildings on streets may lead to resource over-distribution or traffic congestion, thus reducing vitality; whereas in more developed cities like Guangzhou, the number of buildings may have reached saturation, and street vitality is more influenced by other factors, such as building quality, service facilities, and public space configuration.

Figure 6A, representing Haikou's local effect plot for diversity, shows a positive correlation between social function mix and street vitality. The SHAP value increases with the mix from 0 to approximately 1.5, indicating a beneficial contribution to street vitality within this range. Around a social function mix of 1.5, the SHAP value peaks and then slightly decreases, suggesting that beyond this point, additional mixing may have a diminishing effect on vitality. Figure 6B, representing Nanning, displays a similar increasing trend in SHAP values with social function mix up to about 1.5, positively contributing to street vitality. However, the decline in SHAP value beyond 1.5 is more pronounced in Nanning, indicating that excessive mixing may have a more significant negative impact on street vitality compared to Haikou. In Guangzhou (Figure 6C), the SHAP value for social function mix and street vitality shows a continuous rise from 0, increasing steadily until approximately 2.0. The optimal points for Haikou and Nanning appear around 1.5, while Guangzhou's optimal point seems to be around 2.0, suggesting that the ideal level of social function mix might be influenced by the city's development stage.

From Haikou to Guangzhou, as cities develop, the positive contribution of social function mix to street vitality seems to extend to higher levels of mixing, reflecting that more developed cities' street vitality may benefit from more complex built environments.

Figure 6D (Haikou) observes that as the density of recreational facilities increases within a relatively low range, its contribution to street vitality also increases. However, beyond a certain level, the increase in SHAP value slows down, indicating a diminishing positive impact of recreational facility density on street vitality. In Figure 6E (Nanning), an increase in recreational facility density significantly positively impacts street vitality's SHAP value, showing a pronounced upward trend. After a certain point, further increases in recreational facilities seem to saturate in contribution to street vitality, with a slight decrease in SHAP value. Figure 6F, excluding noise in the data, shows no distinct clustering trend in Guangzhou, with more dispersed data points. This indicates that the contribution of recreational facility density to street vitality in Guangzhou is more complex and varied,

which is logical due to ① socioeconomic diversity: Guangzhou, being a more economically advanced and complex large city, has a diverse socioeconomic background. This means the same recreational facility density might have different impacts in different neighborhoods due to socioeconomic conditions. ② Urban planning and layout: the more intricate urban planning and spatial layout of Guangzhou may result in an uneven impact on street vitality due to the interwoven layout of recreational facilities with other urban functions like commercial, residential, and transportation.

Figure 6G shows that the SHAP value initially increases with residential facility density, then decreases. The peak positive SHAP value occurs at a residential facility density of about 20, suggesting this density optimizes the positive impact on street vitality. As residential facility density continues to increase, the SHAP value diminishes, indicating that excessively high residential facility density may no longer positively influence street vitality. Figure 6H (Nanning) displays a clear trend: the increase in residential facility density correlates proportionally with the increase in street vitality up to about 20. Beyond a density of 20, the SHAP value stabilizes or slightly decreases, suggesting that the contribution of residential facility density to street vitality saturates at a certain level. In Guangzhou's chart (Figure 6I), the SHAP value is higher at lower residential facility densities, decreasing initially with increasing density, then stabilizing.

Figure 6J (Haikou) demonstrates a nonlinear decline in SHAP values with office facility density increasing from 0 to about 50, indicating a gradual decrease in the positive contribution of office facility density to street vitality in this range. Figure 6K (Nanning) shows a wider distribution of SHAP values, with office facility density ranging from 0 to over 100. In Figure 6L (Guangzhou), the range of office facility density expands further, exhibiting broader variations from 0 to about 350. The SHAP values rapidly decline with increasing office facility density, reaching approximately $-0.1$, and stabilize at negative values after exceeding about 50, suggesting that in Guangzhou, a higher level of office facility density may adversely affect street vitality.

Cities in the early stages of development, represented by Haikou, may have fewer high-density office areas. The increase in office facilities might directly boost street vitality, potentially linked to more job opportunities and business activities. Mid-stage development cities like Nanning, possibly in a transitional phase, may experience a more complex relationship between office facility density and street vitality, influenced by various urban development factors. In more advanced cities like Guangzhou, with multiple business centers developed, increased office facility density might not directly enhance street vitality and could even lead to vitality decline in certain areas due to resource over-concentration.

In Haikou's chart (Figure 6M), the SHAP value trends from negative to positive with increasing choice, implying that low choice might negatively impact street vitality, while high choice contributes positively. Nanning's chart (Figure 6N) shows a similar but more pronounced trend. As choice increases, the SHAP value first declines and then rises, indicating a more significant impact of choice on street vitality. In high-choice ranges, increased diversity seems to correlate with increased street vitality. Guangzhou's chart (Figure 6O) is more complex than the previous two cities. Positive contributions of SHAP values are observed at low and high choice intervals, while negative values appear in the mid-range. From Haikou to Guangzhou, the impact of choice on street vitality evolves from a singular trend to a more complex pattern, possibly reflecting a nonlinear relationship between urban development level and street vitality as cities increase in complexity. In early-stage development cities like Haikou, simpler linear patterns might be more common. However, as cities develop and become more complex (as in Guangzhou), multivariate and nonlinear influencing factors begin to play a role, making the relationship between street vitality and choice more variable.

Figure 6P (Haikou) shows concentrated negative SHAP values in a certain range of integration, indicating reduced street vitality. As integration increases, SHAP values shift from negative to positive, suggesting a positive correlation between higher integration and street vitality. Figure 6Q (Nanning) displays a trend similar to Haikou's but with a broader

area of positive SHAP values, indicating a more apparent positive impact of integration on street vitality. The upward trend of SHAP values starts earlier, making the impact of integration more widespread. Figure 6R (Guangzhou) shows a more complex relationship between SHAP values and integration without a simple linear correlation. An increase in integration appears to contribute negatively (negative SHAP values) to street vitality in certain ranges.

It can be observed that from cities in the early stages of development, represented by Haikou, to more developed cities like Guangzhou, there is a positive correlation between street integration and vitality, especially in areas of higher integration. Across the three cities, the positive correlation between building height density and integration strengthens, possibly signifying a higher degree of urban development and concentrated vitality. Street integration is a crucial factor influencing urban vitality, but cities at different stages of development respond differently to this factor. As cities develop, the concentration of vitality on highly integrated streets may increase, and this concentration trend might be more pronounced in more developed cities. This phenomenon could be associated with more concentrated economic activities and higher building densities in highly developed cities. However, it should also be noted that excessively high integration might lead to traffic congestion, subsequently reducing street vitality.

### 5.3. Interaction Effects between the Built Environments of Streets

Figure 7A (Haikou) shows a decrease in interaction values with increasing social function density. The highest building height density (warm tones) is associated with lower social function density values. This suggests that in Haikou, areas with higher social function density may not benefit significantly from high building density, possibly due to saturation or other limiting factors. Figure 7B (Nanning), also predominantly displays negative interaction effect values but with less distribution. Data points indicate a possible moderate negative interaction effect between social density and office facilities. This could imply that in Nanning, an increase in office facilities does not necessarily have a positive impact on social function density, or there might be diminishing returns beyond a certain point. Figure 7C (Guangzhou) shows a wider range of interaction effect values, with a relative shift towards positive in the medium range of social density compared to Nanning. Guangzhou's more developed status may indicate more positive interactions between office facilities and social functions.

During different stages of urban development, the relationship between social function density and other indices of the built environment (such as building height density, office facilities, etc.) may change. Smaller cities like Haikou may show certain synergistic effects between social function density and building height density in the early stages of development, whereas larger cities like Guangzhou might exhibit positive synergistic effects with office facilities over a wider range of social function densities, reflecting a more mature and diverse urban structure and economic activities.

Figure 7D (Haikou) observes a negative correlation between increased building perimeter density and SHAP interaction values at low social function densities, indicating that in areas with lower social function density, an increase in building perimeter density may not be conducive to the development of social functions. At high social function densities, data points are more dispersed, suggesting that in areas with denser social functions, the impact of building perimeter density on social functions becomes more uncertain. For Figure 7E (Nanning), the points in the graph show some degree of negative correlation at both levels of social function density, but in high-density areas, the data points are more compact, possibly indicating that as the city develops, the relationship between building perimeter density and social functions becomes more stable. For Figure 7F (Guangzhou), the range of SHAP values in high-density areas is larger, hinting at a more complex relationship between building perimeter density and social functions at high social function densities. In contrast, the data points in low-density areas are more concentrated, and SHAP values are mostly negative, suggesting that in more developed cities, areas with lower social function

density are more sensitive to increases in building perimeter density. As a city's level of development increases, the impact of social function density on building perimeter density may become more complex. In rapidly developing cities, the variability in high-density areas increases, possibly due to the diversity of functions in these areas and the complexity of the built environment.

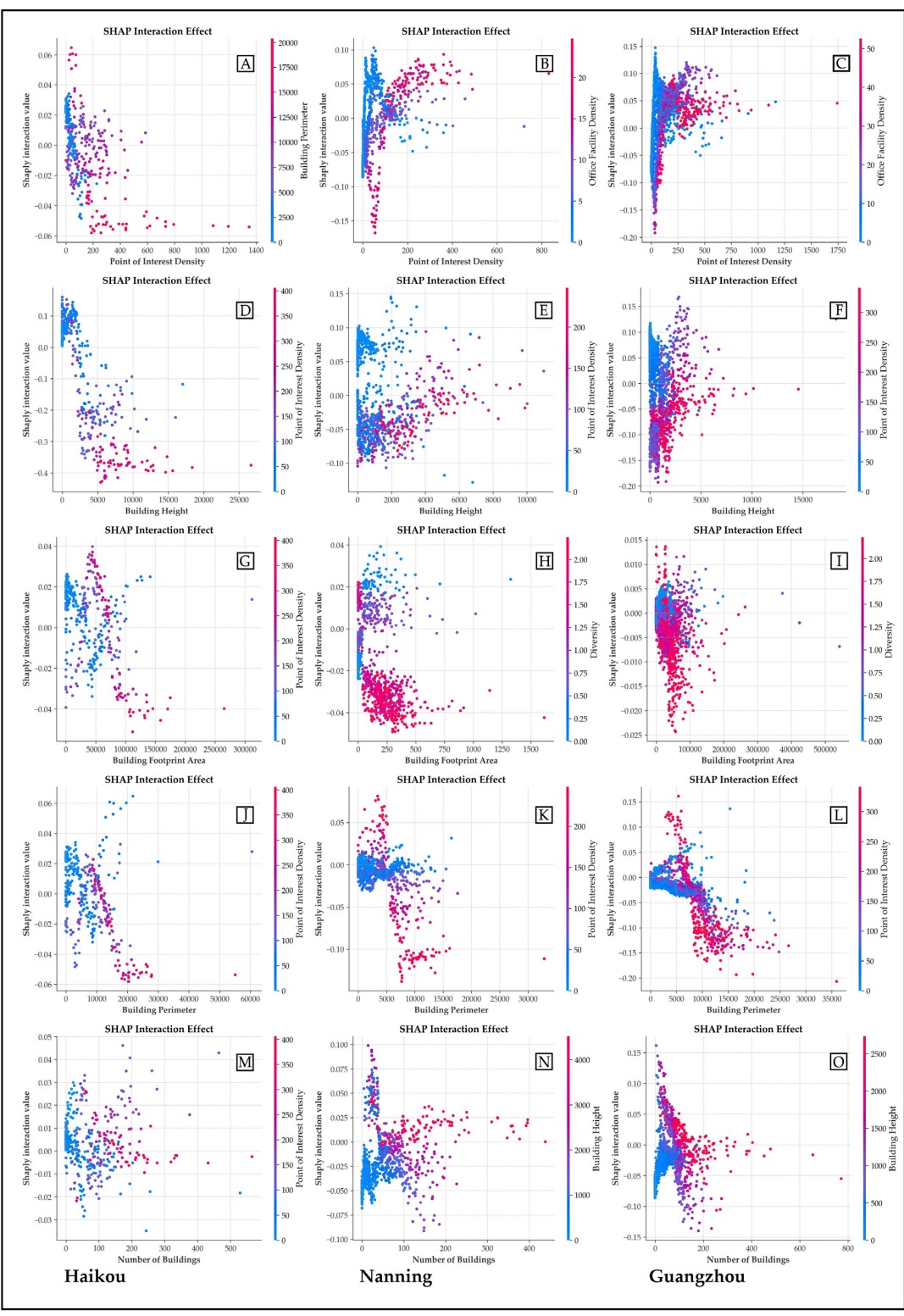

**Figure 7.** Synergies between indicators of the built environment (Indicator 1–Indicator 5).

Figure 7G (Haikou) shows that within the range of low to medium building quantity density, the distribution of blue points indicates a clear positive correlation with social function density. This may represent that in certain areas or types of buildings in the city, an increase in density at a low state can significantly enhance social functions. The red points exhibit a more dispersed relationship, suggesting that these buildings may be located in different areas or have different purposes, and their impact on social function density is less apparent than the blue points. This could be indicative of newer or planned areas where an increase in building quantity density does not directly lead to an enhancement of social functions. In Figure 7H (Nanning), the blue points show a generally negative correlation between building height and building quantity density. This could imply that in some areas, an increase in building height does not bring a relative increase in the number of buildings or that the increase in building height does not have a positive impact on social functions. The red points indicate a more complex but overall negative correlation between building height and building quantity density. This may reflect different planning and development strategies in different areas of Nanning city, where an increase in building height does not directly correspond to an increase in building quantity or enhanced social functions. Figure 7I (Guangzhou) observes more significant negative correlations in the blue points compared to the Nanning graph, suggesting that in some areas, despite a large number of buildings, increased height does not correspond with an enhancement in social functions or that social functions have reached a saturation point. The red points show a reduced negative correlation with building quantity density, indicating that in these areas, increasing building height might help improve social function density or that the relationship between social function and building height is more positive compared to other areas.

As a relatively less developed city, Haikou's building quantity might more significantly impact social functions, as infrastructure and services might not have reached saturation. In more developed cities like Nanning and Guangzhou, there might already be sufficient building infrastructure to support their social functions, so additional buildings may not bring significant improvements in social function. As cities develop economically, the function of buildings may shift from singular to diversified. In economically developed cities like Guangzhou, increasing building height might be to meet diversified commercial and residential needs, rather than just increasing space utilization. In terms of social needs, a city's changing social demands are also reflected in the use and construction of buildings. For example, in densely populated areas, high-rise buildings might be used for residential and office purposes, which may not directly increase building quantity density but might improve social function density.

Figure 7J (Haikou) demonstrates that in areas with lower social function density, as building footprint density increases, SHAP interaction values show a change from positive to negative. This may indicate that in these areas, an increase in building footprint density initially has a positive effect on social functions, but beyond a certain point, it may have a suppressive effect. In areas with high social function density (red regions), SHAP interaction values are generally low, suggesting that in these areas, increasing building footprint density has a weaker or even negative impact on social functions. Figure 7K (Nanning) reveals that in areas with low diversity, SHAP interaction values are negative, and this negative impact increases with the increase in spatial proportion. In areas with high diversity, SHAP interaction values fluctuate around zero, indicating that in these areas, an increase in building footprint density may have both positive and negative effects on diversity, though this relationship is not as apparent as in areas with low diversity. Figure 7L (Guangzhou) shows, similar to Nanning, that SHAP interaction values in low-diversity areas are generally negative and seem more significant as spatial proportion increases. Unlike Nanning, the distribution of SHAP interaction values in high-diversity areas is broader, suggesting that in more developed cities, an increase in building footprint density in high-diversity areas might cause more complex impacts, with both positive and negative effects becoming more evident.

Social function density and diversity, these two indicators, may play different roles in different stages of a city's development. In the early stages of development, an increase in social function density might have a more pronounced positive effect, along with an increase in building density. In more maturely developed cities, diversity might become a more significant factor, influencing the city's spatial layout and economic activities in more complex ways.

In the early stages of urban development, an increase in building footprint density might positively impact social function density. However, as the city further develops and spaces become more compact, this impact might turn negative, especially when reaching a certain saturation level.

Figure 7M (Haikou) shows that from lower to medium building height density data points, SHAP values change from positive to near zero. This might indicate that in Haikou, when building height density is at a lower to medium level, its positive relationship with social function density is stronger. Figure 7N (Nanning) observes that red data points are mainly concentrated in areas with high building height density, with mostly negative SHAP values. This group of data points covers from low to high building height density, with a more dispersed distribution of SHAP values. This suggests that in Nanning, the impact of building height density on social function density might not be as apparent as in Haikou but is more complex and diversified. The fewer red data points in Nanning, mainly occurring in areas with high building height density, show negative SHAP values. Although not numerous, this still indicates that very high building density might have a negative impact on social function density to some extent. Figure 7O (Guangzhou) shows SHAP values ranging from positive to negative over a wide coverage, displaying a more chaotic trend than the previous two cities. This might reflect that in more maturely developed cities, the impact of building height density on social function density could be obscured by other factors or is nonlinear. Although there are more red data points with generally negative SHAP values in areas of high building height density, this might suggest that in developed cities, very high building density has a more apparent negative impact on social functionality, possibly due to overcrowding, reduced green spaces, and public areas caused by excessive building density.

Regardless of the stage of urban development, there appears to be a common characteristic that excessively high building height density might have a negative impact on social functions. This is clearly manifested in the red data points across various graphs. The above trends prompt urban planners to consider different strategies at different stages of development. In the early stage, moderate increases in building height and density could enhance social functions. At the intermediate and advanced stages, more attention should be paid to the quality and multi-functionality of urban space, ensuring that an increase in building density does not negatively impact living quality and social interactions.

Figure 8A (Haikou) illustrates that as social function density increases (color shifting from light to dark red), the positive synergy with diversity seems more pronounced. The blue points are more dispersed, but a relative majority are below the zero line, indicating that in areas with lower social function density, an increase in diversity may relate to negative synergies. Figure 8B, representing Nanning's diversity interaction effect analysis, shows red points relatively concentrated above the zero line, similar to Haikou, suggesting that diversity usually has a positive synergistic effect in areas with high social function density. The blue points are scattered, roughly evenly distributed above and below the zero line, indicating inconsistent impacts of diversity in areas with low social function density, both positive and negative. Figure 8C for Guangzhou demonstrates that the red points, primarily above the zero line, also exhibit a positive synergistic effect of diversity in areas with high social function density. As diversity increases, the red points show an upward trend, indicating that the positive synergistic effect strengthens with increased diversity. The blue points, although dispersed, show more apparent negative synergistic effects compared to Nanning and Haikou, suggesting that in areas with low social function density, an increase in diversity might bring more negative impacts.

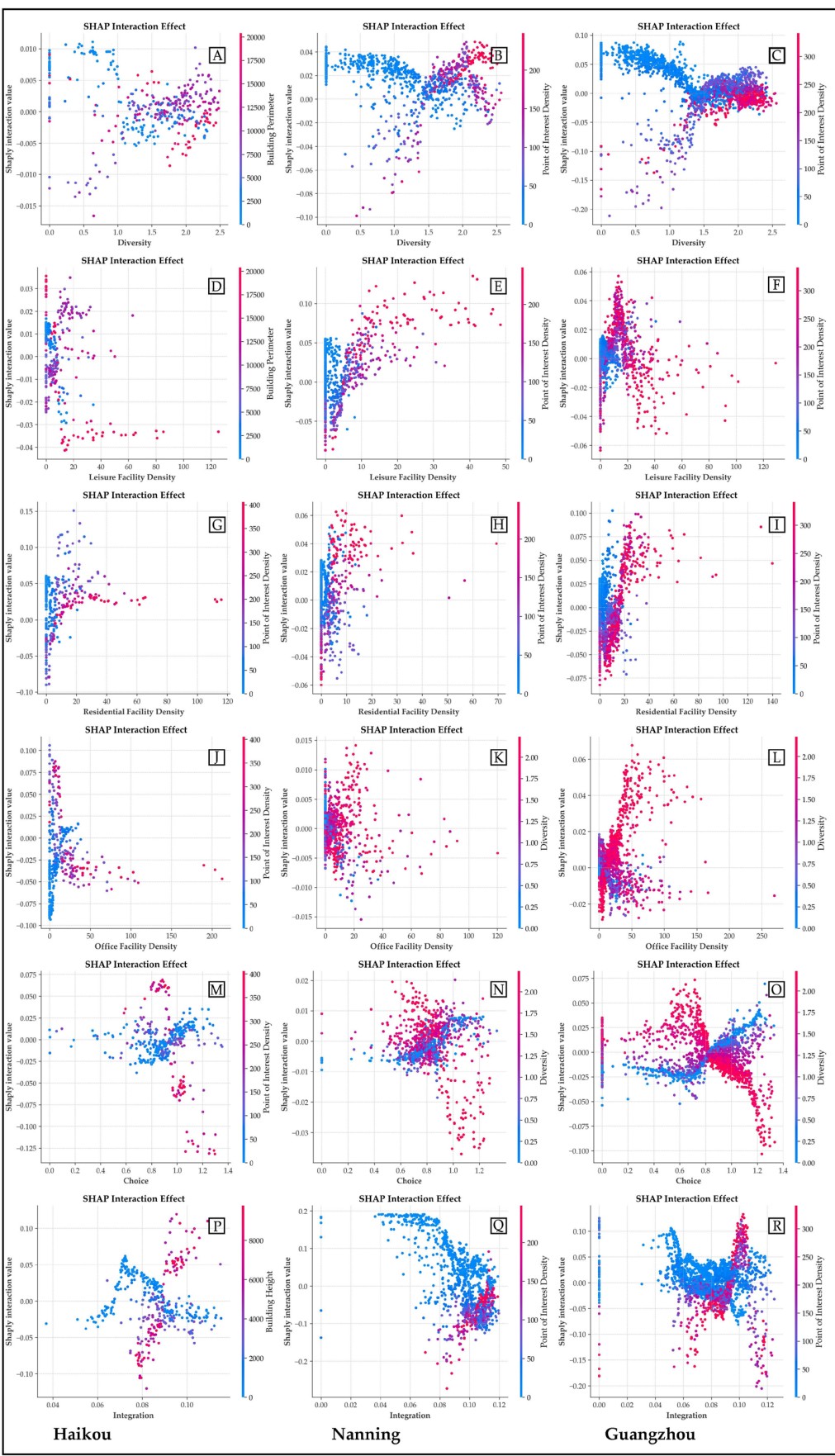

**Figure 8.** Synergies between indicators of the built environment (indicators 6–11).

This could mean that cities in their early development stages, like Haikou, should first enhance social function density before gradually increasing diversity. Mid-stage development cities like Nanning need a more balanced approach to developing diversity, simultaneously enhancing the positive impacts of social function and diversity. For cities with higher social function density, like Guangzhou, further increasing diversity could bring greater economic and social benefits.

Figure 8D (Haikou) shows the synergistic effect of leisure facilities and social function density. When leisure facility density is low, the blue points' interaction effect is near zero or slightly negative. As leisure facility density increases, the blue points exhibit a slight positive growth in interaction effect. This suggests that in areas with lower building density, increasing leisure facilities might help improve the environment's attractiveness. For the red points, the interaction effect seems to decrease with increased leisure facility density. This could indicate that in areas already having high building density, additional leisure facilities no longer significantly enhance environmental quality, or a saturation point has been reached. Figure 8E (Nanning) demonstrates that an increase in leisure facility density appears to have a significant positive impact in areas with low social density, with interaction effects increasing with leisure facility density. This could suggest that in sparsely populated areas, new leisure facilities help attract residents and social activities, strengthening social ties. However, in areas with high social density, the positive impact of additional leisure facilities seems to decrease. This might indicate that in areas already bustling with social activities, new leisure facilities do not significantly enhance social density. Figure 8F (Guangzhou) shows a positive interaction effect in low social density areas with increased leisure facility density, although this effect is less apparent than in Nanning. This might imply that in a more developed city, increasing leisure facilities in low-density areas still positively affect social activities. In high social density areas, the interaction effect seems little changed, or there is a slight decrease, indicating that in already dense social activity areas, new leisure facilities do not significantly increase social interaction.

The data from the three cities suggest that the addition of leisure facilities generally has a positive synergistic effect in low-density areas (both in terms of building and social density). In high-density areas, this synergistic effect either diminishes or becomes less significant. This could indicate that leisure facilities, as a form of social infrastructure, are more effective in enhancing the attractiveness of low-density areas, while their marginal benefits may decrease in high-density areas. For planners, this implies that the planning and investment of leisure facilities should focus more on lower-density areas, especially where the population or buildings are not very dense. In these areas, new or improved leisure facilities could bring greater social and economic returns. For areas that are already maturely developed with high social and building density, the direct benefits of adding leisure facilities might be smaller. Innovative or targeted solutions might be needed in these areas to further improve living quality and social interaction. This also reveals a potential trend that as cities develop and densities increase, the impact of different types of urban infrastructure (such as leisure facilities) on the urban environment changes. In the early stages of development, the positive effects of increasing infrastructure might be more noticeable. In the later stages of urban development, these effects might reach saturation or even require more adjustments and optimization to cope with the complexity of the urban environment and the needs of the residents.

Figure 8G (Haikou) shows residential facility density primarily concentrated in the 0–20 range at lower levels, with SHAP interaction values between −0.10 and 0.15, displaying a distinct peak area. Positive SHAP values indicate that an increase in residential facility density correlates positively with social function density to some extent. Figure 8H (Nanning) displays a slightly broader range of residential facility density, still mainly concentrated at lower levels, around 0–30. SHAP interaction values are mainly concentrated between −0.06 and 0.06, with a more dispersed peak area compared to Haikou. The distribution of interaction values is more even than Haikou, with less distinct boundaries

between positive and negative effects. Figure 8I (Guangzhou) shows the broadest range of residential facility density, covering from low to high, reflecting a wider distribution of residential facilities. The dense area of data points is in the SHAP value range of 0.025 to 0.100, showing a clear upward-right trend. The persistence of positive SHAP values at higher residential facility densities, compared to the other two cities, might indicate that in Guangzhou, the increase in residential facility density has a more significant positive impact on social function density.

For early-stage development cities like Haikou, the transition from low-density residential areas to high-density communities might occur, with social function facilities developing alongside population density growth. In mid-stage development areas like Nanning, as the city further develops, a more complex interdependency might emerge between residential and social function facilities. Cities at a moderate development level might experience certain spatial separation, leading to diversified distributions of social function facilities. In more maturely developed cities like Guangzhou, the relationship between residential and social functions might be more stable, with positive correlations between the two maintained over a broader range of facility densities, reflecting more integrated and balanced spatial planning.

Figure 8J (Haikou) presents the synergistic relationship between office facility density and social function density. The data points indicate that in areas with low office facility density (near the origin), an increase in social function density shows a strong positive correlation with an increase in office facilities. However, as office facility density increases, this positive correlation weakens, as reflected by a decrease in SHAP interaction values. Figure 8K (Nanning) explores the relationship between office facilities and diversity. At lower levels of office facility density, there is a certain degree of positive correlation with diversity, although not as pronounced as in Haikou. With the increase in the number of office facilities, the impact on diversity seems to diminish, approaching zero or slightly negative SHAP values. Figure 8L (Guangzhou) examines the relationship between office facilities and diversity. At lower quantities of office facilities, the response of diversity to the increase in office facilities is not apparent. However, with more office facilities, diversity significantly increases, and the growth in SHAP values indicates a strengthening of the positive correlation. This suggests that in Guangzhou, a more mature business area might be present where the increase in office facilities is accompanied by a rise in other types of facilities, such as retail stores, restaurants, or other services, enhancing the area's diversity.

In early-stage development cities like Haikou, office facilities might not be sufficient to attract a wide range of commercial activities. However, in more mature cities like Guangzhou, economic diversification has led to the emergence of more types of office spaces and service facilities, allowing areas with high office facility density to maintain high diversity.

Figure 8M (Haikou) shows that at high social function density (red), with increasing choice, the synergy values mostly cluster around zero. This implies that in early-stage development cities like Haikou, even with high social function density, its synergy with choice does not significantly enhance street vitality. At low social function density (blue), in areas with low choice, the blue points show negative synergy values, suggesting that in environments with low social function density, a reduction in choice negatively impacts street vitality. Figure 8N (Nanning) reveals that at high diversity (red), with increasing choice, there is a slight upward trend in synergy values, indicating that in mid-stage development cities like Nanning, high street diversity coupled with increased choice may promote street vitality. At low diversity (blue), the level of choice seems to have little impact on synergy values, suggesting that in cities at this development stage, when diversity is low, the impact of choice on street vitality is minimal. Figure 8O (Guangzhou) observes that at high diversity (red), the synergy values are highest at a medium level of choice, indicating that for a maturely developed city like Guangzhou, there might be an optimal range of choice under high diversity conditions that maximizes street vitality. At low diversity (blue), low choice corresponds to lower synergy values, indicating that in a well-developed city

like Guangzhou, even if choice increases, street vitality struggles to significantly improve in the presence of low diversity.

For early-stage development cities like Haikou, the focus should be on enhancing social function density, as even with increased choice, street vitality may be limited without sufficient functional support. For mid-stage development cities like Nanning, attention should be given to the synergistic enhancement of diversity and choice, as at this stage, the increased choice can effectively promote street vitality with the support of diversity. For more maturely developed cities like Guangzhou, the strategy should involve finding the optimal balance between choice and diversity to maintain and enhance street vitality. In these cities, excessive increases in choice may not necessarily lead to enhanced street vitality, making the role of diversity particularly important.

Figure 8P (Haikou) observes that at high building density, synergy values seem to be lower in areas with low integration, suggesting that even with high building density, positive street vitality may not be produced in areas lacking integration. As integration increases, synergy values show a rising trend, but the magnitude of the increase is limited, indicating a certain limitation to the positive impact of integration on street vitality. At low building density, increasing integration does not show a clear trend of increase or decrease in synergy values. This might indicate that for a less developed city like Haikou, low building density is insufficient to significantly impact street vitality. Figure 8Q (Nanning) observes that at high building density, synergy values in areas with medium integration are higher than at the extremes, forming an arch-shaped distribution. This suggests the existence of an optimal range of integration where the synergistic effect of social density and street vitality is maximized. At low building density, synergy values seem to be lower in areas with high integration. This could mean that under conditions of low social density, high integration may not be conducive to enhancing street vitality. Figure 8R (Guangzhou) notes that at high building density, synergy values significantly increase with higher integration, indicating that under high social density conditions, the increase in integration has a very apparent positive effect on street vitality. At low building density, very low or high integration corresponds to lower synergy values. This indicates that in a highly developed city like Guangzhou, street vitality in low social-density areas relies more on a moderate level of integration.

For early-stage development cities like Haikou, enhancing urban integration and infrastructure investment may be key to boosting street vitality. Mid-stage development cities like Nanning should find a balance between increasing social density and integration, avoiding excess in either aspect. For more mature international metropolises like Guangzhou, the strategy may involve maintaining vitality on the foundation of high integration and high social density, preventing negative congestion effects.

*5.4. Discussion*

In this study, GBDT and SHAP are used to investigate the nonlinear and synergistic effects of various indicators on street vitality in the built environment of tropical hot and humid cities with different development processes, represented by Haikou, Nanning, and Guangzhou. We quantify the street-built environment in terms of 11 indicators in four dimensions and use multi-source big data to measure street vitality. We developed a GBDT model and used SHAP for local interpretation. The local interpretation can reflect the contribution of each indicator of the built environment at different scales. Based on the local interpretation, we reveal the relative contribution of built environment indicators to street vitality, their nonlinear effects on street vitality, and the synergistic effects among built environment indicators. Finally, we propose targeted development strategies for cities at different stages of development to promote street vitality.

We find that the development level of a city significantly influences the extent to which various factors contribute to street vitality. For example, in cities at the early stages of development, as represented by Haikou, the density of social functions, building perimeters, number of buildings, and building heights have significant positive effects on street

vitality, but these effects tend to be saturated or declining after reaching a certain level. In mid-development cities, as represented by Nanning, the relationship between these factors and street vitality is more complex, showing more changes and uncertainties. In highly developed cities (e.g., Guangzhou), on the other hand, the positive effects of these factors (especially building height density) on street vitality are persistent and widespread, although diminishing marginal effects may occur at very high levels. This has similarities with the conclusions of Shi and others' research, which states that "high density does not always ensure a high-vitality city" [49].

In addition, factors such as the mix of social functions, the density, diversity, and choice of leisure and residential facilities were found to have different effects on street vitality in cities at different stages of development. For example, the mix of social functions and the density of leisure facilities had a positive impact on street vitality in all three cities, but the extent of their impact varied with the level of urban development. In more maturely developed cities (e.g., Guangzhou), the positive contribution of these factors to street vitality can be sustained to higher levels.

Overall, this study reveals the important influence of the level of urban development on street vitality and the different factors that need to be considered in urban planning and management for cities at different stages of development. Cities in the early stages of development, need to focus on the enhancement of social function density and architectural features while maintaining a moderate development density to avoid negative impacts on street vitality. For more maturely developed cities, there is a need to focus more on the enhancement of diversity, choice, and integration and how to maintain and enhance street vitality in the midst of high-density development. These findings provide valuable guidance to urban planners in formulating appropriate strategies at different stages of development to promote the continued growth and sustainability of urban street vitality.

The local dependency plot shows that there is a general nonlinear effect of the indicators of the built environment on street vitality, aligning with the research and hypotheses of Tao, Liu, Hatami, and others [50–52], with one or more inflection points (thresholds) in the values of all the variables, and their local effects shift from negative to positive or from positive to negative. The rate of increase or decrease in the local effect often changes before and after the inflection point. For example, the rise in local effects is accelerated when the diversity of Canton exceeds 1.5, when the value of the contribution of diversity to street vitality increases. There are also two types of inflection points: upper and lower limits. The former, such as POI density, has a saturation value; the latter, such as POI density in Guangzhou, starts to turn positive around 30. These nonlinear and threshold effects can provide city managers with subtle knowledge to improve street vitality within an effective range.

The effect of one indicator of the built environment can be amplified or diminished as another indicator changes. For example, when the built intensity exceeds about 100, the synergistic effect of the two is then enhanced, and the localized effect of the two is amplified as the diversity of streets increases. Therefore, if the planning objective is to promote street vitality, then it is recommended to prioritize the development of street building intensity and then pursue street diversity on this basis.

The research by Huang, Fu, and Guo, along with Jacobs' classic theory, particularly emphasizes the crucial role of diversity in enhancing street vitality. However, they did not consider whether diversity still holds absolute importance in cities at an early stage of development with lower construction intensity [13,18,46]. Our study finds that although diversity is integral throughout various stages of urban development, construction intensity is more prominent than diversity in the initial stages of urban growth. Haikou, Nanning, and Guangzhou, while sharing similar geographical and climatic conditions, still exhibit differences in their levels of economic development. The majority of built environment indicators influence street vitality in a similar trend, yet they have distinct thresholds and turning points. Policies should be tailored to the development stage of the city, catering to cities at different levels of economic progress. For example, for cities in early development

stages like Haikou, the focus should be on enhancing social functional density dimensions, such as building perimeter density, building number density, and building height density. Key indicators like building height and number density should be monitored to ensure they do not exceed levels that negatively impact street vitality. For cities in mid-development stages like Nanning, the impact of diversity on vitality, particularly in complex and dynamic urban contexts, should be considered. The mix of social functions should be assessed to ensure an appropriate density and diversity of leisure and residential facilities. For highly developed cities like Guangzhou, local effect maps should be used to identify and monitor key turning points or thresholds, such as diversity, POI density, etc., to formulate more detailed strategies around these thresholds.

## 6. Summary and Future Work

This study also raises several issues that warrant exploration in the future. Firstly, due to the absence of a precise definition of street vitality, further efforts should be made in this regard. For instance, this research utilized Baidu Heat Map data and LBSs data as external representations of street vitality while overlooking the involvement of specific social groups, such as non-smartphone users (including individuals without Baidu-related apps installed on their phones), in the computation of street vitality. Secondly, urban streets vary in width, and during the study, a uniform buffer was applied for statistical purposes across different street grades, thereby neglecting the distinct boundary characteristics of various streets. Furthermore, as elucidated in Agnieszka Starzyk's research, the vibrancy of streets or theaters is not solely determined by how individuals interact with spaces but also by the frequency and depth of these interactions. In essence, this vitality arises from the mutual influence and interaction between people and the built environment [53]. However, this study treats humid subtropical regions as homogeneous entities, aiming to investigate the general impact of the built environment on street vitality, thus overlooking the spatio-temporal diversity and heterogeneity within different urban street spaces in the region. Additionally, while this study empirically demonstrated the nonlinear relationship between the built environment of humid subtropical regions and the formation of street vitality, it remains unknown whether similar patterns exist in cities located in other climatic zones. This aspect represents a gap in the research. Nonetheless, developing countries urgently require measures to promote street vitality, and in-depth exploration of the theoretical and practical relationships between the built environment and street vitality holds significant importance.

**Author Contributions:** Conceptualization, J.L. and S.L.; Methodology, J.L. and S.L.; Software, J.L.; Validation, J.L. and S.L.; Formal analysis, J.L.; Investigation, J.L., N.K., Y.K., J.Z. and J.C.; Resources, J.L. and S.L.; Data curation, J.L. and S.L.; Writing—original draft, J.L.; Writing—review and editing, J.L. and S.L.; Project administration, S.L.; Funding acquisition, S.L. All authors have read and agreed to the published version of the manuscript.

**Funding:** This research was funded by the National Natural Science Foundation of China, grant number "52268011" and "The APC was funded by Hainan University".

**Institutional Review Board Statement:** Not applicable.

**Informed Consent Statement:** Not applicable.

**Data Availability Statement:** The data presented in this study are available on request from the corresponding author.

**Acknowledgments:** We are grateful for the assistance from Hainan University.

**Conflicts of Interest:** The authors declare no conflicts of interest.

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
