# Peer review of "Nonlinear and Synergistic Effects of Built Environment Indicators on Street Vitality: A Case Study of Humid and Hot Urban Cities"

_sustainability, doi:10.3390/su16051731_

Round 1

Reviewer 1 Report

Comments and Suggestions for Authors

Basically, the intention of the article is extremely interesting and also purposeful and it will certainly be exciting to see what else will be worked on or published in this field in the future. For this reason, a large part of the comments on this article are of a formal nature and less focused on content.

1. Firstly, attention should be paid to the correct template and the publisher's format specifications. For example, the template from 2021 is used and some other specifications, e.g. line indentation, are not taken into account.

2. the English text is well written and formulated, but there are repeated typos. These should definitely be corrected, including the shift in capitalisation in Table 1.

3. all tables should be revised and adapted to the format of the template. Quite often the words are spread over several lines because the appropriate size was not taken into account.

4. the table should also be further adapted so that a clearer distinction can be made between the contents of the columns and rows. It is sometimes very difficult to read the tables when the information merges into one another.

5. the quality of the illustration should definitely be improved. Some of the labelling, e.g. in Figure 3, is not legible. Figures 5 to 8 are also sometimes difficult to read and, as with the tables, the information in the centre column is not separated clearly enough from the information in the right and left columns.

6 In the course of the text, it is not clear why the authors so clearly refer to humid and hot cities in the title, but this fact is not really taken up or explained again at any time in the text itself. What is the intention behind this? Do the authors want to draw a comparison with cities or countries with different temperatures? Do the authors want to show that the climate also has an influence on urban vitality? None of this is answered.

7 The question also arises as to how the authors define humid or hot in Chapter 4.1. when presenting the study areas.

8 The same applies to the question of how the three cities are categorised. What parameters are used to determine that the cities are differently developed? How are early development, mid-development etc. defined? This should definitely be explained for outsiders.

9) Figure 1 also needs a qualitative improvement.

10. at the end of the presentation, a comparison should definitely be made with the findings from other studies on this topic. Where are there similarities, where do the results differ from other studies, what lessons can we learn from them?

Comments on the Quality of English Language

The English text is well written and formulated, but there are repeated typos.

Author Response

Thank you very much for your valuable review comments on this paper. Please review the attached revisions. Wishing you a pleasant day.

Reviewer 2 Report

Comments and Suggestions for Authors

Dear Author(s),

Thank you for this paper. I appreciated your work. Anyway, before publication I suggest you some review your text as follows:

Section 4, Line 186: Please, could you explain, even in a few lines "why" you chose to examine these cities?

Figure 1: Could you provide a bigger map of the cities? The figure is too small.

Finally, please, review the entire text to avoid some typos.

Author Response

(The authors gave the same response as above.)

Reviewer 3 Report

Comments and Suggestions for Authors

The manuscript is not well prepared following the journal guideline. Too much editing or typing errors throughout, including line space, font, reference format. The whole content is far from informative and inductive. Please read the guide for authors, and strictly use the template with all necessary sections, such as funding information and all statements. Moreover, edit all equations in the right format. Misuse or inconsistency in the reference list. In all, please re-structure the whole stuff and re-submit with necessary language polishing service.  

Comments on the Quality of English Language

Extensive editing of English language required

Author Response

(The authors gave the same response as above.)

Round 2

Reviewer 3 Report

Comments and Suggestions for Authors

Authors have re-formatted the whole stuff following the Journal instructions. The case modelling and mapping-based analysis do make some senses. The main problem for the present form lies in the distance from scientific value and presentation.

1. Abstract is general, without specific information. Key quantitative statements are necessary.

2. Re-organize the whole content in a more scientific way. The present form is more like a case report, rather than an academic article.

3. Re-structure the manuscript with sperate sections in Introduction, Materials/Method, Results etc. order. The literature review should be incorporated into backgrounds or intro part. Split the modelling results or case studies into stratified results and discussion.

4. What is the main research gap and the objective. Clarify the main study focus based on the background.

5. Clarify the novelty of this paper. What are the key new findings? The used method and analysis approach are all off-shelf handy ones. What makes the chosen cities typical or representative cases? 

6. Compare the results or conceptions with other similar studies in other cities worldwide. How about the key advance or progress over available research. 

7. Comparative study or benchmark investigation on climatic conditions? How do meteorological parameters impact the key specific results?

8. What is the key research meaning? Reference value for other cities? Why chosen case is typical? Hypothesis and modelling validation? Sensitivity analysis for the discussed various influences or considerations? Without these details or information, the findings or recommendations seems easily to be meaningless.

9. Still several editing and formatting errors or typos throughout. All equations should be numbered consecutively.   

Comments on the Quality of English Language

 Extensive editing of English language required.

Author Response

Once again, I extend my heartfelt gratitude for your guidance. The process of revising my paper has been incredibly educational, highlighting several research aspects that I had either not learned previously or had not paid sufficient attention to. First and foremost, I have gained a deeper understanding of the importance of formatting in Word, which I will be sure to pay close attention to in future submissions. Secondly, it has underscored the need to enhance my knowledge and application of mathematical statistics. Lastly, it has brought to light the importance of extensive literature review and adhering to scientific writing standards for paper composition. In sum, your teachings will be of lifelong benefit to me. I am immensely grateful for the time you took to provide invaluable suggestions for my article. I wish you success in your work and happiness in your life.

Round 3

Reviewer 3 Report

Comments and Suggestions for Authors

The authors have attempted to address each comment, improving the structure and presentation of their work. While several responses unsatisfactory, I believe the manuscript meets the minimum requirements for consideration for publication. But the authors need extensive language polish work and professional format editing, since many inconsistent expressions (see authors affiliation? 1 or 2 the same? Figure and Table caption font). 

Comments on the Quality of English Language

Extensive editing of English language required.

Author Response

Thank you once again for your valuable comments on the manuscript, and for your recognition of the paper. We have used the official MDPI proofreading service to polish the manuscript, including some adjustments to the format. Regarding the authors' affiliations, we have made corrections in accordance with the submission guidelines.

The Chinese Lunar New Year is coming soon, I wish you smooth work, a happy life, and a joyous holiday.
